# Zinc-indium-sulfide favors efficient C − H bond activation by concerted proton-coupled electron transfer

Xuejiao Wu [1,3] ✉, Xueting Fan[2,3], Shunji Xie[2], Ivan Scodeller[1], Xiaojian Wen[2], Dario Vangestel [1], Jun Cheng [2] ✉ & Bert Sels [1] ✉

C − H bond activation is a ubiquitous reaction that remains a major challenge in chemistry. Although semiconductor-based photocatalysis is promising, the C − H bond activation mechanism remains elusive. Herein, we report value-added coupling products from a wide variety of biomass and fossil-derived reagents, formed via C − H bond activation over zinc-indium-sulfides (Zn-In-S). Contrary to the commonly accepted stepwise electron-proton transfer pathway (PE-ET) for semiconductors, our experimental and theoretical studies evidence a concerted proton-coupled electron transfer (CPET) pathway. A pioneering microkinetic study, considering the relevant elementary steps of the surface chemistry, reveals a faster C − H activation with Zn-In-S because of circumventing formation of a charged radical, as it happens in PE-ET where it retards the catalysis due to strong site adsorption. For CPET over Zn-In-S, H abstraction, forming a neutral radical, is rate-limiting, but having lower energy barriers than that of PE-ET. The rate expressions derived from the microkinetics provide guidelines to rationally design semiconductor catalysis, e.g., for C − H activation, that is based on the CPET mechanism.

C − H activation being among the most omnipresent processes is a key elementary reaction for the valorization of hydrocarbon resources such as fossil and biomass[1–4]. C − H bonds typically possess high bond dissociation energies (BDE) rendering them unreactive, and therefore challenging[1–7]. While efforts for C − H bond chemical activation are in progress[1–7], nature has long found its way to oxidize such C − H bonds efficiently, and this pathway runs via a concerted proton-coupled electron transfer (CPET) mechanism[8]. In numerous biological processes, including the well-known oxidative respiration, photosynthesis, and nitrogen fixation, electron transfer (ET) and proton transfer (PT) often occur in a single concerted step[8]. Such CPET overall enables a decrease of the energy barrier for element−H (X − H) bond activation, ultimately resulting in high reaction rates[8–11].

Photocatalytic activation of C − H bonds has recently emerged as a powerful tool towards novel chemical transformations[12].

Mechanistically, these reactions most likely start by the transfer of light-induced electrons, followed by the proton transfer (ET-PT), also called a stepwise proton-coupled electron transfer (PCET) mechanism[12]. For example, benzylic sp[3] C − H bonds were reported to be activated by ET-PT through arene radical cation intermediates (Fig. 1, route 1)[13–15]. Photocatalysts with highly oxidative excited states, e.g. acridiniums[14,15] and iminiums[13], with $E^{*}_{red} > +2\,V$ vs. the saturated calomel electrode (SCE), are required for catalyzing the ET (first) step, generating the high-energy arene radical cation intermediates. Photocatalysis incorporating hydrogen atom transfer (HAT) has also been explored to get access to such benzylic/allylic radicals. Photo-excited organics, such as aromatic ketones and xanthene dyes, can function as HAT reagents to activate the C − H bonds directly[16]. Alternatively, photo-redox catalysis drives the conversion of organics to corresponding radicals, e.g., thiols to

[1]Center for Sustainable Catalysis and Engineering, Faculty of Bioscience Engineering, KU Leuven, Heverlee 3001, Belgium. [2]College of Chemistry and Chemical Engineering, Xiamen University, Xiamen 361005, China. [3]These authors contributed equally: Xuejiao Wu, Xueting Fan.
✉e-mail: xuejiao.wu@kuleuven.be; chengjun@xmu.edu.cn; bert.sels@kuleuven.be

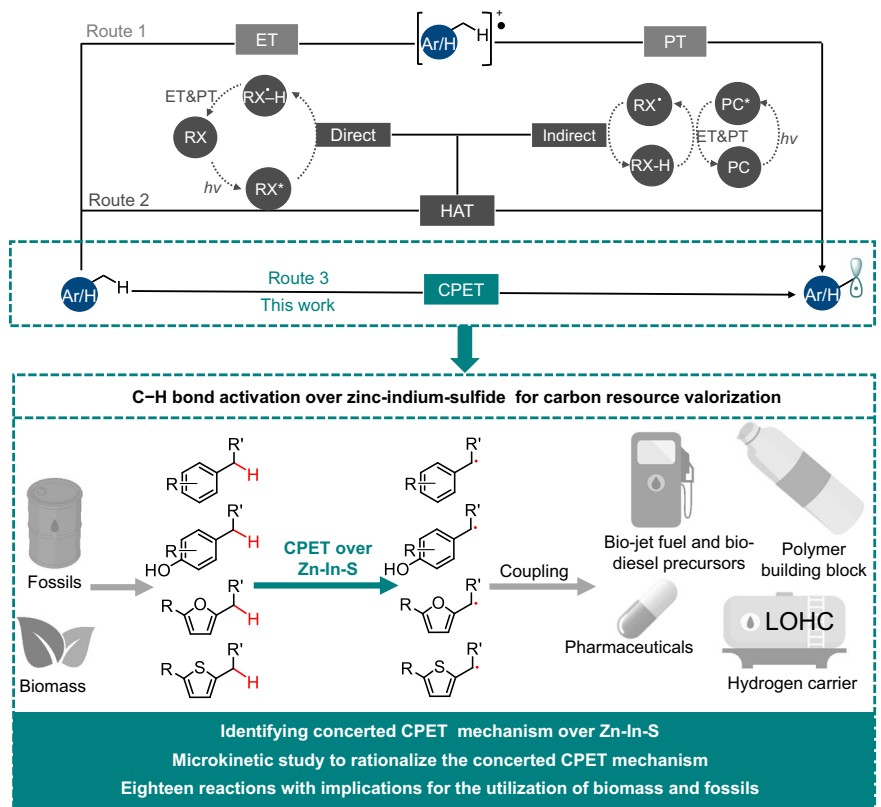

**Fig. 1 | Schematic illustration for key progress in this work for concerted CPET activation of C − H bond by Zn-In-S considering the valorization of fossil- and biomass-derived chemicals.** Ar/H represents aromatics and heterocycles, including furans and thiophenes.

sulfur and amines to nitrogen radicals, which could facilitate C − H bond activation. In this manner, photocatalysis engages in HAT through an indirect approach (Fig. 1, route 2)[17]. More recently, CPET C − H activation has been reported in several photocatalytic systems, which typically consist of homogeneous photocatalysts such as ruthenium complexes, fluorescein, and cyanoarene[18,19]. However, when it comes to the valorization of complex feedstock such as biomass, heterogeneous catalysts can offer several advantages over homogeneous ones, such as better compatibility with complicated chemical environments, higher stability over time, and ease of separation from the organic products[3].

The semiconductor photocatalysis for C − H bond activation has gained a steady interest and substantial advancements over the past decade, particularly in the valorization of biomass has been reported[3,20–23]. Unfortunately, the C − H bond activation mechanism over semiconductors generally remains elusive. While most research assumes ET-PT (Fig. 1, route 1) or HAT, this is concluded without solid experimental evidence[3]. Discerning the general PCET mechanisms into either the concerted CPET or stepwise ET-PT and PT-ET can provide crucial insights that are necessary to rationalize the amelioration of the C − H bond activation efficiency. Current understanding of PCET is majorly derived from model photoactive system studies, in which PCET reagents are homogeneous metal complexes or organic bases[8–11]. Conventional approaches that analyze PCET kinetics in homogeneous systems, i.e., by extracting rate constants through kinetic data fitting, are hard to adopt in heterogeneously catalyzed systems (semiconductors) due to the complex surface chemistry[8–11]. Studies covering interfacial CPET for photocatalytic C − H activation are therefore rare. It is for instance unclear how the driving force and surface properties of the semiconductor alter PCET photocatalysis.

In this study, we present a thorough kinetic and mechanistic investigation to clarify the underlying chemistry of interfacial C – H activation over Zn-In-S. A combination of experimental, viz. isotope, kinetics, and control experiments of reactants with varying C − H BDE, and computational results demonstrates the occurrence of the (semiconductor-rare) CPET mechanism for Zn-In-S, contrasting the otherwise commonly accepted stepwise sequential ET-PT mechanism. Given the complexity associated with photocatalysis on surfaces, we introduced and pioneered microkinetic models, that are based on the relevant elementary steps and then simplified after evaluation using experimental and computational kinetic data. This gave access to rate expressions (and their kinetic parameters) that can guide the design of better Zn-In-S catalysts, e.g. through compositional modifications. For instance, in the presence of the modified Zn-In-S semiconductors, high yield conversion of various hydrocarbons by $C_{sp3}$−$C_{sp3}$, $C_{sp3}$−$C_{sp2}$, and $C_{sp3}$ − N coupling of radical intermediates, following CPET, were achieved (18 examples). The reaction selection creates versatile (self and cross) coupling products, such as precursors of bio-jet fuels, biodiesel, polymer building blocks, liquid organic hydrogen carriers (LOHC), and pharmaceuticals (Fig. 1) among others.

## Results
### Superior performances of ZnIn2S4 for photocatalytic C − H activation
TiO2[24] and Zn-In-S[25] are both popular semiconductors with wide energy and environmental applications. Photocatalytic C − H bond activation in toluene, an abundant chemical from petroleum[26], was examined first over both commercial TiO2, i.e., Degussa P25, and synthesized $ZnIn_2S_4$ (Supplementary Fig. 1). $ZnIn_2S_4$ and P25 possess a similar surface area, viz. 62 and 52 m²/g respectively (Supplementary Table 1). Bibenzyl, methylated dimers, and trimers, because of the oxidative coupling of toluene, were the main products (Supplementary Fig. 2). Interestingly, a 17 times higher coupling rate was observed over $ZnIn_2S_4$, viz. 1.3 vs.

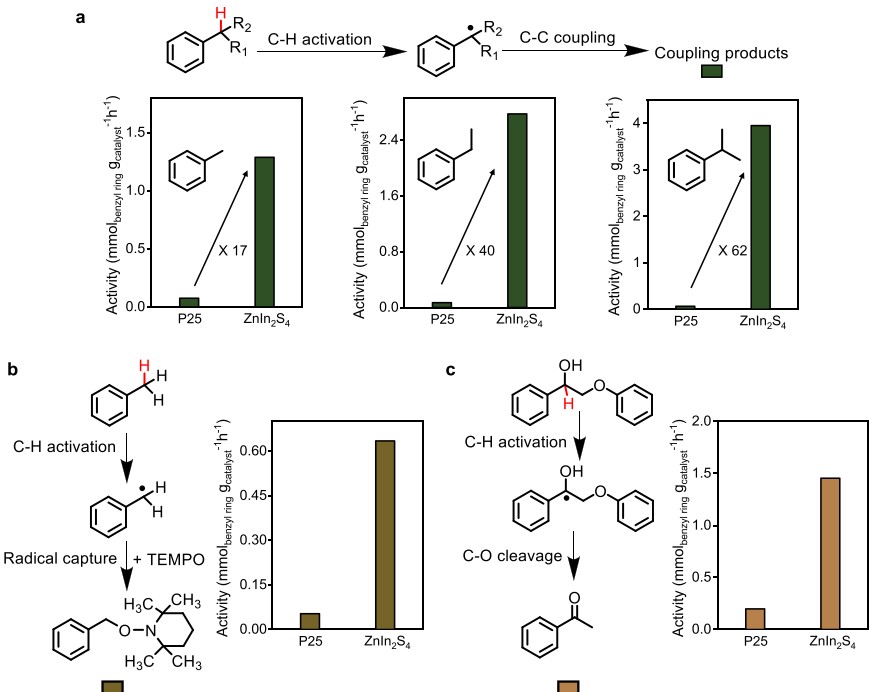

**Fig. 2 | Superior performances of ZnIn₂S₄ for C − H activation. a** Conversion of benzylic chemicals, *i.e.*, toluene, ethylbenzene, and cumene, over P25 and ZnIn₂S₄. **b** The TEMPO-benzyl radical adduct formed over P25 and ZnIn₂S₄. **c** The cleavage of

the C − O bond in 2-phenoxy-1-phenylethanol, which goes through a benzylic radical intermediate from C − H bond activation, over P25 and ZnIn₂S₄.

0.076 mmol/g/h (for P25) (Fig. 2a). H₂, which is known as renewable fuel[27], was also detected in considerable amounts (Supplementary Table 2). Analysis of the electron-hole balance (Supplementary Table 2) by stoichiometric analysis indicates consumption of the photogenerated electrons (from both ZnIn₂S₄ and P25) for H₂ production, while the photogenerated holes drove the oxidative coupling of toluene.

Suppression of the toluene coupling by addition of a small amount of radical-trapping reagent, 2,2,6,6-tetramethylpiperidine-1-oxyl (TEMPO), suggests the involvement of radical intermediates (Supplementary Fig. 3). The mass and ¹H NMR spectrum of the TEMPO-radical adduct indeed indicate formation of the benzyl radical intermediate (Supplementary Fig. 4), which is formed by a C − H activation route, and that is more than ten times faster over ZnIn₂S₄ compared to P25 (Fig. 2b). The high efficiency of ZnIn₂S₄ in driving the photocatalytic C − H activation to the benzyl radical generation is key to its high photocatalytic activity for toluene conversion.

The catalytic efficiency of ZnIn₂S₄ was further evaluated for the conversion of other benzylic chemicals and benchmarked to the results with P25. The activities of ZnIn₂S₄ for ethyl benzene (2.8 mmol/g/h) and cumene (3.95 mmol/g/h) coupling were impressively high; 40 and 62 times higher than that over P25, respectively (Fig. 2a). Besides oxidative coupling of benzylics, the photocatalytic cleavage of the C − O bond in 2-phenoxy-1-phenylethanol, a typical lignin model, was also investigated. Significantly more acetophenone product, generated through the formation of a benzyl radical intermediate by C − H activation[21], was formed over ZnIn₂S₄, as compared to P25 (Fig. 2c).

It is noteworthy to mention that the well-known P25 photocatalyst has been considered a benchmark in different hole-induced systems due to its strong oxidation capacity. On the one hand, the lower valence band maximum (VBM) of P25, viz. 2.53 and 1.78 V (for ZnIn₂S₄) vs. SCE (Supplementary Fig. 5 and Supplementary Fig. 6) indicates stronger oxidation ability of the photo-generated holes for P25, and therefore the higher activity of ZnIn₂S₄ for the oxidative C − H activation of various benzyl chemicals is unexpected (Fig. 2). On the other

hand, P25 exhibited higher activities for photocatalytic oxidations following ET mechanism, such as the reactions with triethylamine and decomposition of bisphenol A (Supplementary Fig. 7). Given ET is the rate-controlling step for the ET-PT reaction due to its high energy barrier, the photocatalytic efficiency usually correlates well with the ET activity[9]. These two observations suggest that the photocatalytic C − H activation over ZnIn₂S₄ does not follow the commonly accepted ET-PT mechanism for this type of semiconductors.

## What is the C − H activation mechanism of Zn-In-S?

The apparent kinetic isotope effect (KIE$_{app}$), i.e., the ratio of coupling rates between toluene and its deuterated forms $d_8$ and $d_3$, over ZnIn₂S₄ was measured. The KIE$_{app}$ values of both were above two, proving a rate determining H transfer, and the values were similar, viz. 2.13 (Fig. 3a) and 2.15 (Supplementary Fig. 8), indicating very limiting β secondary effect contribution. In addition, similar activities were observed in the solvent of CH₃CN and CD₃CN (Supplementary Fig. 9), indicating the influence of solvent composition on the vibrational modes and solvent-solute coupling[28] only has a negligible contribution to KIE. The value above two is thus diagnostic of a primary KIE that is ascribed to CPET or PT-ET[9]. Given toluene is a very weak acid (with pK$_a$ ≈ 41)[29], its deprotonation requires a strong Brønsted base. As ZnIn₂S₄ is not, the involvement of PT-ET is most unlikely, leaving CPET as the elect mechanism. Given its unity KIE$_{app}$ value (Fig. 3a), P25 C − H activation is without any rate-determining H transfer, endorsing ET-PT.

To further support the distinct mechanisms, photocatalytic conversion of several benzyl reagents with different BDEs was examined. The activity of ZnIn₂S₄ increased significantly with decreasing BDEs (Fig. 3b)[30], in agreement to the well-known negative correlation between CPET activity and BDE[31]. In contrast, reaction rates for the different benzyl reagents were similar over P25 (Fig. 3b). Due to the analogous alkyl chain functionalized benzyl ring in these reagents, their oxidation potentials are very similar[32], giving rise to similar ET rates and thus comparable ET-PT photocatalytic activity in

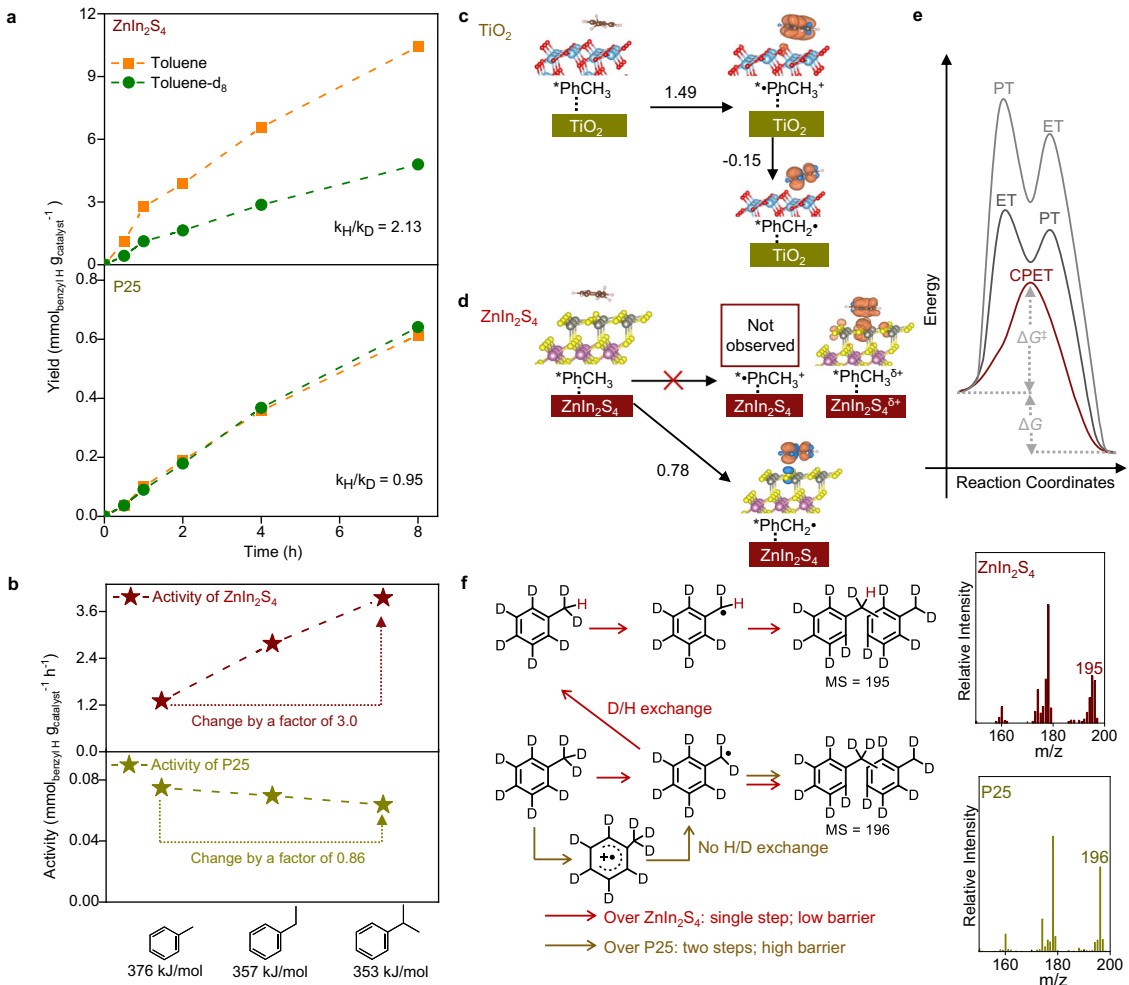

**Fig. 3 | Mechanistic insights for C−H activation over ZnIn₂S₄ and P25.**
**a** Apparent kinetic isotopic studies for photocatalytic conversion of toluene and toluene-$d_8$ over ZnIn₂S₄ and P25. **b**, Dependence of coupling rates over ZnIn₂S₄ and P25 on benzylic reagents with different C−H bond dissociation energy (BDE). **c, d** Spin densities and energetics of the proton and electron transfer steps were calculated with the PBE functional on (**c**), the anatase TiO₂ (101) surface, and, (**d**), the hexagonal ZnIn₂S₄ (001) surface. In (**c, d**), the spin densities of the hole are visualized by orange *iso*-surfaces. The light blue, red, grey, pink, yellow, brown, and white balls represent Ti, O, Zn, In, S, C, and H atoms, respectively. The energies are in eV. **e** Conceptual illustration of the energetic advantages of the CPET process. $\Delta G$ and $\Delta G^{\ddagger}$ represent free energy change and activation barrier for the CPET process, respectively. **f** Mass spectra of methylated dimers from deuterated toluene-$d_8$ over P25 and ZnIn₂S₄, as well as the proposed mechanism.

the presence of P25. This result strongly supports CPET C−H activation mechanism for ZnIn₂S₄, as opposed to ET-PT for P25[33].

We performed density functional theory (DFT) calculations for the toluene oxidative coupling on both the anatase TiO₂(101) and hexagonal ZnIn₂S₄(001) surface. We first optimized the initial structure including a hole on the TiO₂(101) surface. Interestingly, the hole is located on the toluene molecule (Fig. 3c, TiO₂-*PhCH₃⁺, in which * indicates surface adsorbed), suggesting that the photo-induced hole can oxidize *PhCH₃ directly. The oxidation potential is calculated to be 1.49 V (vs. SCE; Fig. 3c), which is less positive than the VBM position (2.53 V vs. SCE) of TiO₂. Since this confirms the capability of the photoexcited hole to oxidize toluene on the TiO₂ surface, a two-step ET-PT route via •PhCH₃⁺ to PhCH₂• in the case of TiO₂ is most likely. In contrast, ZnIn₂S₄ has a less positive VBM position (1.78 V vs. SCE), while the hole is mainly delocalized on the ZnIn₂S₄ surface and no stable oxidation intermediate product with the hole localized on the PhCH₃ molecule is observed (Fig. 3d, with the formation of ZnIn₂S₄$^{\delta+}$-*PhCH₃$^{\delta+}$ while ZnIn₂S₄-*PhCH₃⁺ can not be found). This indicates a CPET-based route to form PhCH₂• on the surface of ZnIn₂S₄. The calculated dehydrogenation potential (0.78 V vs. SCE) (Fig. 3d) is less positive than the VBM position of ZnIn₂S₄, suggesting that the oxidative

dehydrogenation of PhCH₃ to PhCH₂• on ZnIn₂S₄ is thermodynamically highly favorable.

The CPET mechanism (in the case of ZnIn₂S₄ semiconductor) may thus be key for its high efficiency in C−H bond activation and coupling product formation. For benzylic C−H activation in toluene, the high 2.26 V oxidation potential renders the ET-PT pathway difficult to drive, and toluene is a very weak acid (pKₐ in CH₃CN = 41), requiring a strong base for the PT-ET pathway. On the other hand, deprotonation of the toluene cation radical (PT in ET-PT pathway, pKₐ = −13) and oxidation of toluene anion radical (ET in PT-ET pathway, E = −0.926), the second step for both sequential pathways, are highly thermodynamically favorable (Supplementary Fig. 10)[13]. Thus, coupling the favorable event (second step) through the concerted pathway is then able to compensate for unfavorable energetics (first step). The much more favorable driving forces in the concerted approach are often accompanied by low activation barriers, allowing concerted CPET to proceed more rapidly than competing sequential transfer pathways (Fig. 3e)[9]. Besides, it is known that concerted mechanisms such as CPET can possess kinetic advantages giving a favorable driving force that can decrease the activation barrier (as illustrated in Fig. 3e) for the rate-determining C−H activation step.

This allows a faster C − H bond activation compared to a sequential pathway such as ET-PT and PT-ET[34]. The C − H rate enhancement can be evidenced by detailed inspection of the different product features from the photocatalytic conversion of deuterated toluene-$d_8$ over $ZnIn_2S_4$ and P25. Multiple molecular ion peaks, ranging from 193 to 196 m/z, were observed in the methylated dimers fraction obtained over $ZnIn_2S_4$. Fragment masses 193 to 195 m/z are methylated dimers from a single to multiple D/H exchange. This is consistent with the reversibility of the CPET C − H bond activation, contributing to the one-step and low reaction barrier characteristics (Fig. 3f)[35]. Oppositely, no such D/H exchange was observed over P25, where the major molecular mass, viz. m/z 196, is attributed to the fully deuterated methylated dimer, and this can only be explained by the occurrence of ET-PT, which is a two-step process with higher reaction barrier (Fig. 3f).

## Pioneering microkinetic analysis of CPET and ET-PT mechanisms over semiconductors

The general form of the rate expression for photocatalysis is[36]:

$$\text{Rate}(r) = k \times \theta \times (e_s^- \text{ or } h_s^+)^n \tag{1}$$

where $k$ is the kinetic constant of the rate determine step (RDS), $\theta$ is the photocatalyst surface that is occupied by the target reagent, and $e_s^-$ or $h_s^+$ represent surface charge carriers. Value for $e_s^-$ or $h_s^+$ is determined by the light irradiation set-up and intrinsic properties of the photocatalyst. These factors collectively control the generation, diffusion, trapping, and recombination of $e^-$ and $h^+$ (Electron part in Fig. 4a). The value of exponent $n$ is determined by the surface elementary steps and their kinetic relevance, and in most cases, this exponent represents the order of the reactions.

To develop microkinetic models for CPET and ET-PT over semiconductors, we have enumerated essential elementary steps involved in both mechanisms (Fig. 4b, c). The complex reaction network for the oxidative toluene coupling over semiconductors complicates a straightforward mechanistic understanding. Therefore, we first analyzed and evaluated the kinetic relevance of the elementary steps based on experimental and computational results (as detailed in the text below). By ultimately formulating the more simplified rate expressions, we hope to develop valuable insights (from the kinetic parameters) that can guide the design of optimal photocatalysts with high activity. Note that our analytical expressions are formulated based on assumptions that surface sites for adsorption and reaction are identical, and interactions between adsorbed species are negligible. The occupation of surface sites by charge carriers and $H^+$ is disregarded due to their small volume compared to $PhCH_3$ and its surface intermediates.

For Zn-In-S, the interfacial reaction of toluene (Fig. 4b) starts with its adsorption on the surface (A1), followed by CPET C − H activation to produce the surface adsorbed $\cdot PhCH_2$ (A2) and a proton. The reactive $\cdot PhCH_2$ radical, which has an open-shell electronic structure, can easily receive electrons from the semiconductor and undergo predominant reverse reactions, if it is binding on the surface[13,37]. In other words, for $\cdot PhCH_2$ to participate in the subsequent coupling reactions, it needs to be desorbed from the surface to the near-surface or bulk solvent region (A3). Meanwhile, photo-generated electrons drive the reduction of the protons to the formation of $H_2$ (A4). Free $\cdot PhCH_2$ in the solution undergoes C − C coupling (A5) to produce the $PhCH_2 − CH_2Ph$ couplings product.

The adsorption energies of $PhCH_3$ and $\cdot PhCH_2$ to the surface of $ZnIn_2S_4$ were estimated by DFT calculation (Supplementary Table 3). It was found that the adsorption of $PhCH_3$ is considerably more favorable than the $\cdot PhCH_2$ desorption, ruling out the possibility of A1 as RDS. Additionally, the adsorption of $\cdot PhCH_2$ is also weak and mainly by the Van der Waals interaction with the surface (Supplementary Table 3

and Supplementary Fig. 11), indicating a low desorption energy barrier. Therefore, the desorption of $\cdot PhCH_2$ (A3) is unlikely to be RDS and can thus be considered at equilibrium. Dibenzyl ketone, which can be decomposed under UV light irradiation to alternatively generate similar benzyl radicals very quickly[38], is employed as the substrate to investigate its coupling rate in the presence of the solid catalyst (A5). Given the coupling rates were at least 25 times higher than the C − H activation rates (Supplementary Fig. 12), A5 cannot be RDS. Based on these analyses, we can deduce that the production of $\cdot PhCH_2$ by the CPET C − H activation (A2) or the extraction of $e_s^-$ to produce $H_2$ (A4) most likely limits the overall rate.

Assuming A2 is rate-limiting, the coupling product generation rate by the CPET mechanism over Zn-In-S can be described by the following rate expression (see SI for details):

$$r_{Zn-In-S} = k_{CPET} \times \frac{K_{Ad1} \times c_{RC-H}}{2 \times (K_{Ad1} \times c_{RC-H} + K_{Ad2} \times c_{RC\bullet} + 1)} \times h_s^+ \tag{2}$$

On the other hand, when A4 is considered RDS, the following rate expression can be derived (see SI for details):

$$r_{Zn-In-S} = k_R \times \frac{(K_{CPET} \times K_{Ad1} \times c_{RC-H})^2}{(K_{Ad2} \times c_{RC\bullet})^2} \times (h_s^+ \times e_s^-)^2 \tag{3}$$

Given the photocatalytic properties of Zn-In-S can easily be modified by compositional changes[39], more information can be gained by investigating the compositional effect on the photocatalytic kinetics. We synthesized two additional compositions, viz. $Zn_{0.5}In_2S_{3.5}$ and $Zn_2In_2S_5$, and tested both for toluene coupling. The reaction rates were calculated in mmol/m²/h to account for variations in surface area (Supplementary Table 1). The reaction rates were found to increase upon the increase of the Zn ratio (Fig. 4d).

When $H_2$ evolution is rate-determining (A4), as illustrated in Eq. (3), a fourth power dependency of activity to the surface electrons ($h_s^+ \approx e_s^-$) is expected. In this case, the activity should be mainly determined by the intrinsic properties of Zn-In-S, namely, its efficiency in light absorption, charge separation, and charge escaping. Transient photocurrent responses, which are proportional to $\eta_{\text{light adsorption}} \times \eta_{\text{charge separation}} \times \eta_{\text{charge escaping}}$ ($\eta$: efficiency)[40], were measured to evaluate the electronic properties of catalysts. The photocurrent intensity increased in the order of $Zn_{0.5}In_2S_{3.5} < Zn_2In_2S_5 < ZnIn_2S_4$ (Supplementary Fig. 13), which is different from the increased order of the observed C − H activation activity. Additionally, the activity of $Zn_2In_2S_5$ demonstrates a linear increase with the rise in toluene concentration from 2.5 vol% to 10 vol%, with a slight deceleration from 10 vol% to 20 vol% (Supplementary Fig. 14). With A4 as RDS, a two-order dependency on the toluene concentration would be expected. The observed first-order dependency confirms that A2 determines the rate. According to Eq. (2), under the condition of small adsorption energy (Supplementary Table 3), a linear relation of activity to toluene concentration is anticipated. These results indicate that the oxidative half-reaction (A2), rather than the reductive half-reaction (A4), is RDS. This is consistent with most metal sulfide-based hydrogen evolution systems, in which oxidation by photo-generated holes is slower than $H_2$ generation by the photo-generated electrons[41].

In the case of A2 as RDS, Eq. (2), the values of $K_{Adx}$ for Zn-In-S of different compositions are similar due to the almost identical surface structures of the Zn-In-S series[39], and this is confirmed by the comparable adsorption energies of $PhCH_3$ and $PhCH_2\bullet$ on $ZnIn_2S_4$ and $Zn_2In_2S_5$ (DFT calculations; Supplementary Table 3). Variations in the activity should thus be attributed to changes in $k_{CPET}$, i.e., the $\cdot PhCH_2$ radical formation upon surface H abstraction, which can be expressed

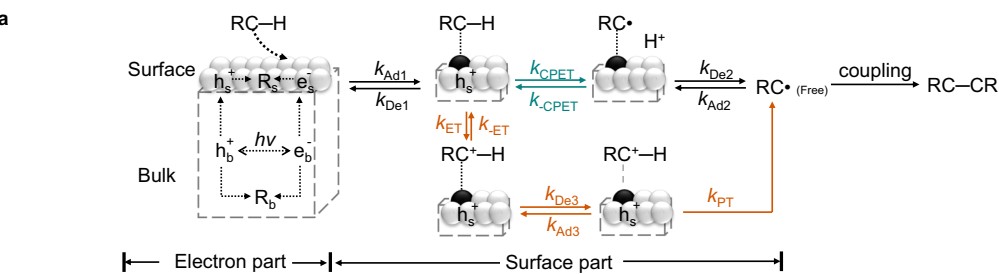

**a**

**b** CPET over Zn-In-S:

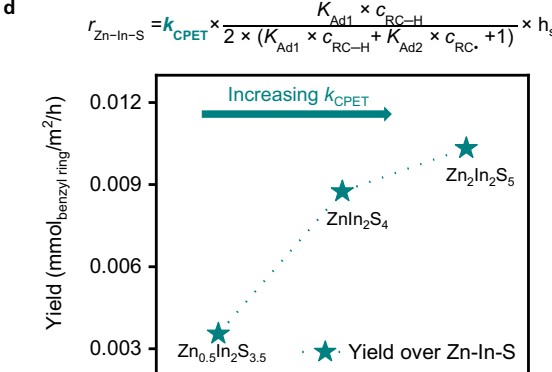

**c** ET-PT over TiO$_2$:

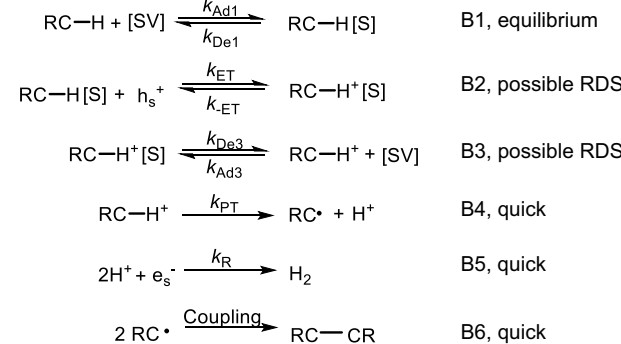

**d**

$$r_{Zn-In-S} = k_{CPET} \times \frac{K_{Ad1} \times c_{RC-H}}{2 \times (K_{Ad1} \times c_{RC-H} + K_{Ad2} \times c_{RC\bullet} + 1)} \times h_s^+$$

**e**

$$r_{TiO2} = k_{De3} \times \frac{K_{ET} \times K_{Ad1} \times c_{RC-H}}{2 \times (K_{Ad1} \times c_{RC-H} + K_{ET} \times h_s^+ \times K_{Ad1} \times c_{RC-H} + 1)} \times h_s^+$$

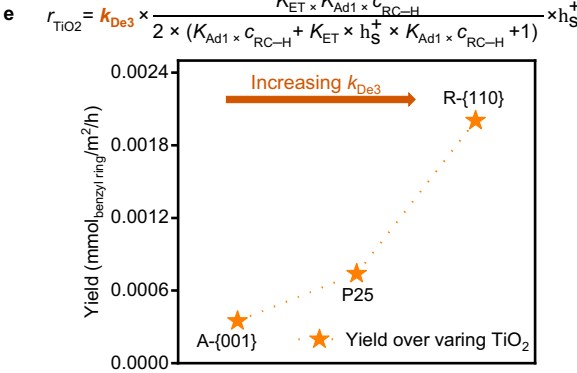

**Fig. 4 | Microkinetic analysis for C − H activation by CPET mechanism over Zn-In-S and by ET-PT mechanism over TiO$_2$. a** Graphical depiction of elementary reactions involved in the oxidative coupling of toluene. Chemical formulas of elementary steps involved in the photocatalytic coupling of toluene over (**b**), Zn-In-S and (**c**), TiO$_2$. $^a$The kinetic relevance of elementary steps, i.e., equilibrium, quick reaction, and possible RDS, is analyzed in the main text. [AS] represents available surface sites, while 'X'[S] represents 'X' species-occupied surface sites. $^b$For elementary steps at their equilibrium, the forward reaction rate equals the reverse reaction rate, and the equilibrium constant ($K_x$) equals to the forward reaction rate constant divided by the reverse reaction rate constant. **d** Increasing CPET rates by changing Zn-In-S composition to increase $k_{CPET}$. **e** Increasing ET-PT rates by changing TiO$_2$ exposed facets to increase $k_{De3}$. The rates in (**d**, **e**) are expressed in mmol/m$^2$/h to account for variations in surface area.

by the Marcus-type formulation ($0 < -\Delta G_{\mu,v} < \lambda$)[42]:

$$k_{CPET} = \sum_{\mu,v} \frac{P_\mu}{h} |V^{el} S_{\mu,v}|^2 \sqrt{\frac{\pi}{\lambda k_B T}} \exp\left[\frac{-(\Delta G_{\mu,v} + \lambda)^2}{4\lambda k_B T}\right] \quad (4)$$

in which the exponential function part indicates a strong kinetic dependency of $k_{CPET}$ on the thermodynamic driving force ($\Delta G_{\mu,v}$). $\Delta G_{\mu,v}$ is the sum of $\Delta G_{ET}$ and $\Delta G_{PT}$, and $\Delta G_{ET}$ increases with increasing Zn-to-In ratio, leading to the larger VBMs, viz. changing from 1.72 to 1.83 V for Zn$_{0.5}$In$_2$S$_{3.5}$ and Zn$_2$In$_2$S$_5$ (Supplementary Fig. 15), respectively. Therefore, the observed activity increase of Zn-In-S with different compositions (Fig. 4d) is most likely ascribed to the increase of $k_{CPET}$ contributing to the downward shift of VBMs. Based on these findings, it can be inferred that A2, viz. formation of weakly adsorbed

•PhCH$_2$, is RDS for the photocatalytic C − H activation of toluene over Zn-In-S.

In the case of ET-PT over TiO$_2$ (Fig. 4c), a reaction network like CPET over Zn-In-S is proposed, with the main difference being the replacement of CPET C − H activation (A2) into ET-PT C − H activation (B2 and B4). For similar reasons as illustrated above for Zn-In-S, the adsorption of toluene (B1) and the C − C coupling step (B6) are excluded as being rate controlling. Because no isotopic effect was observed for P25 (Fig. 3a), the reduction of H$_2$ (B5) cannot determine the rate. The very low pK$_a$ of toluene radical cation ($\sim$ −8) indicates that its deprotonation (B4) cannot be RDS either[43]. For ET-PT reactions, ET is much slower than PT, which suggests B2 as RDS[9,43]. Unfortunately, the adsorption energy of •PhCH$_3^+$ cannot be calculated. This is because periodic boundary conditions are applied in the modeling, the reference (zero) of the electrostatic potential in such a setup is artificial and

has no physical meaning[44–46]. Addition or removal of the •PhCH$_3^+$ in the model changes the net charge of the cell and therefore also the electrostatic potential reference. Total energies from these calculations are hence useless, and cannot be used to determine relative energies. However, the high reported energy of •PhCH$_3^+$[47] indicates a strong desorption energy. Therefore, B3 for the desorption of radical intermediates cannot be excluded as RDS.

Assuming first B2 is rate controlling, the product formation rate by ET-PT mechanism can be described by the following rate expression (see SI for details):

$$r_{TiO2} = k_{ET} \times \frac{k_{Ad1} \times c_{RC-H}}{2 \times (K_{Ad1} \times C_{RC-H} + K_{Ad3} \times C_{RC-H+} + 1)} \times h_s^+ \quad (5)$$

Assuming otherwise B3 is RDS, the following rate expression can be derived (see SI for details):

$$r_{TiO2} = k_{De3} \times \frac{k_{ET} \times K_{Ad1} \times c_{RC-H}}{2 \times (K_{Ad1} \times C_{RC-H} + K_{ET} \times h_s^+ \times K_{Ad1} \times c_{RC-H} + 1)} \times h_s^+ \quad (6)$$

The structure-activity relationship of different TiO$_2$ catalysts was investigated to gain a more profound comprehension of the photocatalytic kinetics. Anatase TiO$_2$ dominated by high-energy {001} facets (denoted as A-{001}, Supplementary Fig. 16), and rutile TiO$_2$ enclosed mainly by low-energy {110} facets (denoted as R-{110}, Supplementary Fig. 17)[48] were therefore synthesized and tested for the photocatalytic conversion of toluene (Fig. 4e). The order of photocurrent intensity enhancement for different TiO$_2$ catalysts was observed as R-{110} < A-{001} < P25 (Supplementary Fig. 18). Their surface energy increases in the order of R-{110} < P25 < A-{001}[48], and their VBMs also increase in the same order, transitioning from 2.49 V of R-{110} to 2.53 V of P25 to 2.59 V of A-{001} (Supplementary Fig. 19). Interestingly, the photocatalytic toluene coupling activity increased in the order of A-{001} < P25 < R-{110} (Fig. 4e), which is opposite to the trend observed for the surface energy and VBMs. The activity of P25 increases almost linearly with the concentration of toluene to around 20 vol% (Supplementary Fig. 20). This indicates that the denominator part including "c$_{RC-H}$" is negligible and can be ignored, implying a first-order dependency on h$_s^+$ for both Eq. (5) and Eq. (6).

The increase of VBMs contributes to the increase of $k_{ET}$ in Eq. (5), with B2 as RDS (in most photocatalytic systems, it counts that $0 < -\Delta G_{ET} < \lambda$)[9], with

$$k_{ET} = \frac{|V^{el}|^2}{\hbar} \sqrt{\frac{\pi}{\lambda k_B T}} \exp\left[\frac{-(\Delta G_{ET} + \lambda)^2}{4\lambda k_B T}\right] \quad (7)$$

which cannot explain the opposite trend of activity increase.

On the other hand, the decrease of surface energy results in the decrease of desorption energy barrier ($\Delta E_{De}$) and thus the exponential increase of $k_{De}$.

$$k_{De} = A \times \exp(-\frac{\Delta E_{De}}{RT}) \quad (8)$$

Therefore, the increase of activity in the series of TiO$_2$ can only be explained in Eq. (6) by the increasing value of $k_{De3}$. This indicates that the desorption of the radical cation intermediate •PhCH$_3^+$ (according to B3) is most likely RDS for ET-PT over P25 TiO$_2$. Notably, the generation of the radical cation intermediate (from ET-PT), which possesses high adsorption energy on the surface and limits the reaction rate, is avoided in CPET. For ET-PT in homogeneous catalytic systems, ET is normally RDS, while our microkinetic study reveals that the adsorption/desorption of reactive (charged) radical species on the

surface can also determine the kinetics. This is a unique feature that can only be encountered in (surface-based) heterogeneous photocatalytic systems.

## C − H bond activation in fossil- and biomass-derived chemicals

To broaden the scope of chemicals that can be coupled according to the CPET photocatalytic system under visible light irradiation, a series of other (fossil- and biomass-derived) reagents were tested. The most performant catalyst in line with the here proposed CPET mechanism, i.e., with the Zn$_2$In$_2$S$_5$ composition, was employed. In analogy to the photocatalytic conversion of toluene, the conversion of non-phenolic aromatics resulted in straight-chain dimers, methylated dimers, and trimers as major products. High to excellent 72−94% yields of coupling products, potentially interesting as heat transfer fluids and organic hydrogen carriers[49], were achieved under the optimized conditions (Fig. 5a, Supplementary Table 4, and Supplementary Figs. 21−26).

Zn$_2$In$_2$S$_5$ was also used for the conversion of alkyl phenolic chemicals, as representatives of lignin-derived chemicals[50]. Oxidation of phenolic O − H is a competitive reaction to CPET activation of the benzylic C − H, while the use of protonic solvents can hinder this, most likely due to hydrogen bonding with the O − H bonds (Supplementary Fig. 27)[51]. Therefore, solvents with strong hydrogen bonding ability were tested to promote the benzylic C − H bond activation in the alkyl phenolics. As expected, high yields of C − C coupling products (for the models: p-cresol and 4-ethyl-phenol; Supplementary Fig. 28) were observed in solvents with strong H bonding abilities such as in MeOH. Photocatalytic conversion of analogous, but lignin-derived phenolics in MeOH yields 74 to 86% coupling products (Fig. 5b and Supplementary Table 4). Major products include bisphenols, bisguaiacols, and bissyringols, of which the latter two may function as biopolymer precursors or additives, that can substitute the hormone disruptive bisphenols[52,53]. Note that side-products from cross-coupling of phenols and MeOH were also observed (Supplementary Figs. 29−32).

Recently, the coupling of biomass-based furanics was achieved by simple metal sulfides as photocatalysts, while doping with Ru promotes the activity by improving the charge separation efficiency[54]. Their report is consistent with the increasing value of h$_s^+$ in our proposed rate expression, Eq. (2). To further investigate, we tested the conversion of 2-methylfuran over the series of Zn-In-S, and the difference in the value of h$_s^+$ was eliminated by dividing their photocurrent intensities. The normalized activity (mmol/m$^2$/h/h$_s^+$) increased with increasing oxidation potential of the semiconductor (Supplementary Fig. 33), corresponding to higher values $k_{CPET}$ in Eq. (2), showing that the modified Zn$_2$In$_2$S$_5$ is the best photocatalyst also for furan conversion. The experimental results indeed show high yields of coupling products, which can serve as potential precursors for biodiesel and bio-jet fuels (Fig. 5b, Supplementary Table 4, and Supplementary Figs. 34, 35). Additionally, the modified Zn$_2$In$_2$S$_5$ catalyst was utilized for the conversion of thiophenes, substrates that are less explored compared to aromatics and furans, and achieved high coupling product yields ranging from 60% to 92% (Fig. 5b, Supplementary Table 4, and Supplementary Figs. 36−39). This further underscores the versatility of the current CPET-based semiconductor system. Given these results validate the proposed rate equations (and their assumptions), the kinetic equations may be considered as potential guidelines to rationally design better photocatalysts.

In addition to self-coupling, the active CPET photocatalytic system can also be useful for cross-coupling reactions, as illustrated in Fig. 5c. Using an excessive amount of benzyl alcohol as the benzyl source, and nucleophilic 1,3,5-trimethoxybenzene as the coupling partner, 72% yield (based on 1,3,5-trimethoxybenzene) of C$_{sp2}$−C$_{sp3}$ coupling product was obtained (Supplementary Fig. 40). Similarly, with azobenzene as the coupling partner, a 75% yield of the

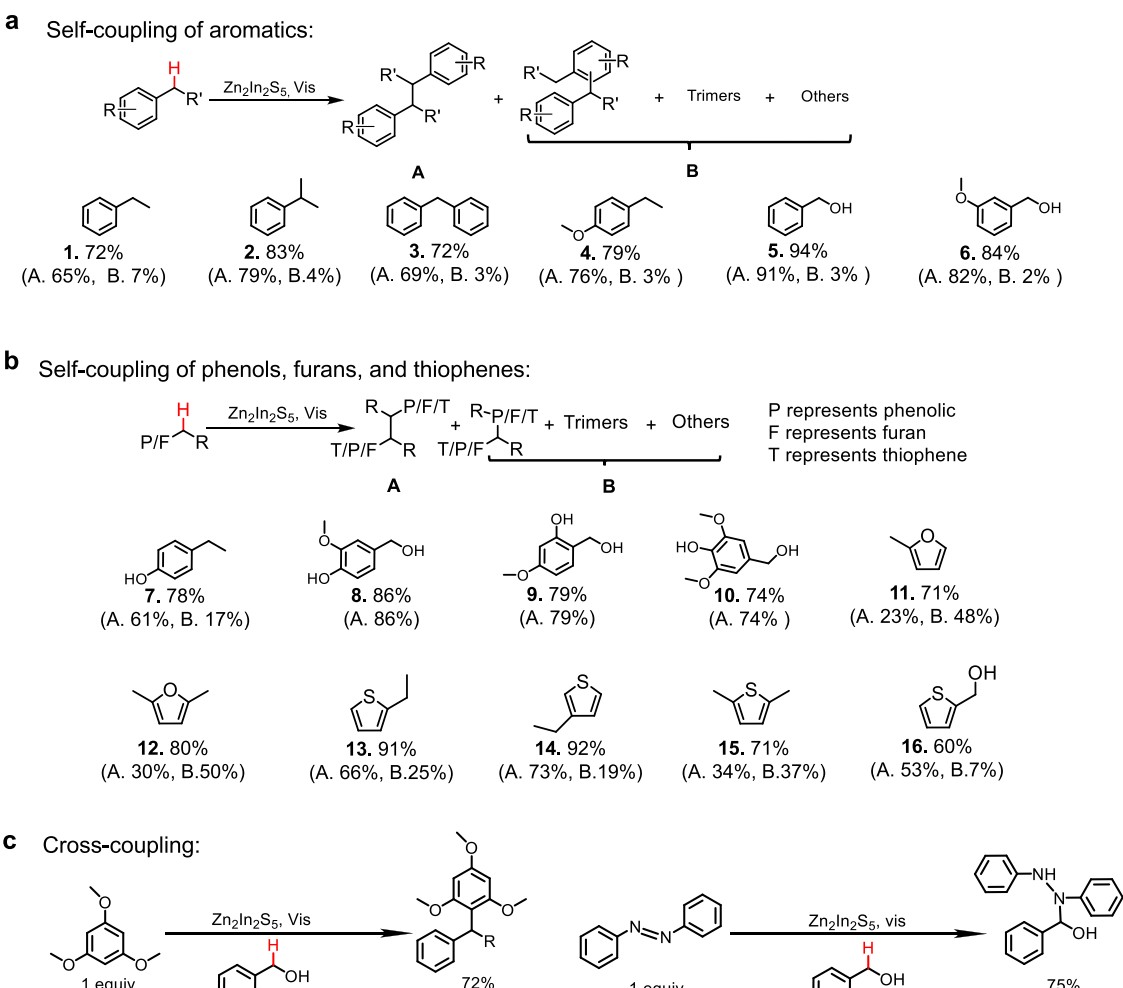

**Fig. 5 | Substrate scope. a** Self-coupling of non-phenolic aromatics. **b** Self-coupling of phenols, furans, and thiophenes. **c** Cross-coupling between benzyl alcohol and chemicals with high nucleophilicity.

corresponding C−N coupling product (based on azobenzene) was achieved (Supplementary Fig. 41). It is worth mentioning that the construction of C−N bonds is of great interest to the pharmaceutical industry, since N-containing compounds are key structural motifs for most current pharmaceuticals. These results demonstrate the feasibility of CPET over Zn-In-S in the efficient photocatalytic valorization of versatile carbon resources.

## Discussion

A variety of fossil and biomass derived chemicals were converted to valuable (self and cross) coupling products with high rates and excellent yields by the Zn-In-S semiconductor. The key to success lies in the high efficiency of Zn-In-S in catalyzing the demanding C−H bond activation to generate a carbon radical intermediate, which is weakly adsorbed on the surface, prior to its coupling. Combined experimental and computational investigations concluded the CPET mechanism, in contrast to the ET-PT mechanism that is commonly accepted for semiconductors such as TiO$_2$. CPET experiences a low activation barrier for radical formation in line with the high C−H bond activation rates. By considering the most relevant elementary steps and evaluating the surface kinetics with both theoretical and experimental data, a microkinetic study (for both CPET over Zn-In-S and ET-PT over P25 TiO$_2$) led to simplified rate expressions with distinct rate-determining steps for CPET and ET-PT. The main learnings are that ET-PT is slow due to the stepwise

formation of a radical cation intermediate that is strongly adsorbed on the surface, whereas formation of the charged radical is circumvented in CPET. Here the concerted transfer of hole and proton with the formation of the (weakly adsorbed) neutral radical intermediate is determining the rate.

Furthermore, rate expressions derived from microkinetic analysis offer insightful guidelines for enhancing the catalytic efficiency in both ET-PT over TiO$_2$ and CPET over Zn-In-S. More specifically, in the case of ET-PT rate expression, Eq. (6), increasing the value of $k_{De3}$ is anticipated to boost activity. Other parameters, including $K_{ET}$, $K_{Ad1}$, and $h_s^+$, are both numerators and denominators of the equations, thereby diminishing their impact on activity. The current study demonstrates the effective enhancement of activity by regulating the exposed facets of TiO$_2$ to decrease $k_{De3}$ and thereby increase activity. On the other hand, for CPET rate expression, Eq. (2), the activity is expected to increase by increasing $h_s^+$ and $k_{CPET}$. While the activity rise by increase of $h_s^+$ has been illustrated in a previous study by another group[54], our work succeeded in increasing $k_{CPET}$ by enhancing $\Delta G_{ET}$ through compositional modifications of Zn-In-S. Note that the increase of $k_{CPET}$ can also be achieved by alternative methods, such as enhancing surface basicity to increase $\Delta G_{PT}$ or altering the solvent to decrease the reorganization energy $\lambda$, Eq. (4), both of which are potential means to increase the photocatalytic activity. It is worth emphasizing that this study represents, to the best of our knowledge, the first application of microkinetics to describe element-H activation chemistry,

revealing the significant role that microkinetics can play in studying the understated yet crucial CPET chemistry in nanoscale interfacial systems.

Overall, the mechanistic insight illustrates that CPET can also be the leading mechanism for C−H activation using semiconductor photocatalysis, while the proposed rate expressions from the microkinetic study offer a nice tool to guide the rational design of more performant C−H activation photocatalytic systems.

## Methods

### Synthesis of Zn-In-S

Metal sulfides, i.e., $Zn_{0.5}In_2S_{3.5}$, $ZnIn_2S_4$, and $Zn_2In_2S_5$ ($Zn_mIn_2S_{m+3}$), were synthesized by a modified low-temperature hydrothermal method[55]. In brief, $ZnCl_2$, and $InCl_3$ ($x$ mmol $ZnCl_2 + 1.5 \times y$ mmol $InCl_3 = 6$ mmol) together with 7.8 mmol of thioacetamide were dissolved in 150 mL of deionized (DI) water. The mixture was heated to 90 °C under vigorous stirring. After maintaining at 90 °C for 5 h under vigorous stirring, the solution was cooled to room temperature naturally. The precipitation was collected by centrifugation, then rinsed with DI water and ethanol, and dried under vacuum at 60 °C overnight.

### Synthesis of TiO$_2$

TiO$_2$, i.e., A-{001} and R-{110}, were synthesized by modified methods[48]. In the synthesis of A-{001}, a mixture of tetrabutyl titanate (25 mL) and 47% hydrofluoric acid solution (3 mL) was heated to 180 °C and maintained at this temperature for 24 h. Subsequently, the mixture was cooled to room temperature, and the resulting sample was separated through centrifugation. The separated sample was then subjected to a series of washes with 1 M NaOH, distilled water, and ethanol, followed by overnight drying at 60 °C. On the other hand, for the synthesis of R-{110}, a mixture of tetrabutyl titanate (10 mL), distilled water (10 mL), and 38% hydrochloric acid (10 mL) was heated to 180 °C and kept at this temperature for 24 h. After cooling to room temperature, the sample was separated via centrifugation and underwent subsequent washing steps with 1 M NaOH, distilled water, and ethanol. The sample was then dried overnight at 60 °C and finally calcined in air at 500 °C for 3 h.

### Evaluation of photocatalytic performance

For the comparison of the photocatalytic activities of commercial Degussa P25 and $ZnIn_2S_4$ in the conversion of toluene and other benzylic chemicals (Figs. 2a, 3a, b), 20 mg catalysts, 10 vol% toluene or other benzylic chemicals in $CH_3CN$ (2 mL) were added into a quartz reactor (10 mL). The reactor was evacuated and purged with Ar for 5 min twice. The reaction mixture was stirred at 850 r.p.m. and irradiated under a 200 W mercury-xenon lamp. For the photocatalytic conversion of 2-phenoxy-1-phenylethanol, triethylamine, and bisphenol A, the reaction conditions were the same as those for the conversion of toluene except for using 2-phenoxy-1-phenylethanol of 0.1 mmol, triethylamine of 0.285 mmol, and bisphenol A of 0.2 mmol as the substrate, and a reaction time of 12 h, 2 h and 8 h, respectively. For different metal sulfides and TiO$_2$ in Fig. 4, an LED light source with 310 nm was used to ensure the responsive light intensity for all catalysts is the same. For the conversion of different fossil- and biomass-derived chemicals (Fig. 5), visible light ($\lambda = 400-780$ nm) was employed to avoid potential side reactions induced by UV light irradiation. The reaction conditions were the same as those for the conversion of toluene in Fig. 2 except for using 10 mg or 10 uL reactant, 10 mg catalyst, and a reaction time of 14 h. Given the lower reactivity of furans and thiophenes, longer reaction times were used to improve the yields of coupling products: 36 h for furans and 24 h for thiophenes. Methanol was used as the solvent for the conversion of phenols. For the decomposition of diphenyl ketone, 20 mg of diphenyl ketone was irradiated for 15 min.

### Calculation of free energies

The free energies ($\Delta G$) of dehydrogenation were calculated using the computational hydrogen electrode (CHE) method[56,57], and the total energy difference is used to estimate the free energy change assuming the entropic contributions are small. The deprotonation energy was calculated using the simplified scheme in which the $pK_a$ difference between surface species and a reference surface site is estimated from the total energy difference of the acid-base reaction on the surface. As required by Hess's Law, the dehydrogenation energy must be equal to the sum of the deprotonation energy and oxidation energy. Therefore, the oxidation energy was obtained by subtracting the deprotonation energy from the dehydrogenation energy. The calculated oxidation energy and dehydrogenation energy were referenced to the standard hydrogen electrode (SHE), and converted into energies vs. the saturated calomel electrode (SCE) by subtracting 0.244 V. To obtain accurate energies, it is important to ensure the calculated intermediates have the correct spin states. Due to the well-known delocalization error at the generalized gradient approximation (GGA) level, the PBE functional may be inadequate to give correct spin states and then the hybrid functional should be used to correctly describe the electronic states of radical intermediates. In this work, we found that the correct spin states have been obtained using the PBE functional (Fig. 3c, d). We also calculated the dehydrogenation free energy of toluene in vacuum using PBE and hybrid HSE06 functional, and both gave very similar results (1.41 eV for PBE and 1.43 eV for HSE06).

### Reporting summary

Further information on research design is available in the Nature Portfolio Reporting Summary linked to this article.

## Data availability

All data supporting the research in this study are available within the article and supplementary information file. Source data are provided with this paper.

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

## Acknowledgements

This work was supported by Research Foundation-Flanders-the FWO (12S1822N, V422221N, and V470023N, all by X. Wu), the National Natural Science Foundation of China (21972115 by S.X., 22022201 by S.X., 22225302 by J.C., 22021001 by J.C., and 21861132015 by J.C.), the China Postdoctoral Science Foundation (Grant Nos. 2020M682079 by X.F.), I.S. and B.S. acknowledges the NEXTbioref project, sponsored by the Flemisch government through the iBOF financing.

## Author contributions

X.W. (Xuejiao Wu) performed most of the experiments, analyzed the data, and co-wrote the paper. X.F. performed most of the DFT computations and analyzed the data. S.X. performed some of the experiments and analyzed the experimental data. I.S. performed the qualification and quantification of some products. X.W. (Xiaojian Wen) performed part of the DFT computations. D.V. co-wrote the paper. J.C. guided the computational work, analyzed all computational data, and co-wrote the paper. B.S. designed and guided the study, and co-wrote the paper. All of the authors have discussed the results and reviewed the manuscript.

## Competing interests

The authors declare no competing interests.
