## [Peer Review File · Nature Communications]

Zinc-indium-sulfide favors efficient C–H bond activation by concerted proton-coupled electron transferREVIEWER COMMENTS

Reviewer #1 (Remarks to the Author):

Semiconductor photocatalysts play a vital role in the valorization of biomass by harnessing light energy to drive chemical reactions. In this manuscript, Cheng, Sels, and co-workers presented a Zn-In-S semiconductor catalyzed C–H bond activation strategy for the transformation of fossil and biomass derived chemicals. The authors carried out a mechanistic investigation to clarify the mechanism of zinc-indium-sulfides interfacial C–H bond activation process through isotope labeling, kinetic studies, and DFT calculations. Meanwhile, a microkinetic study has unveiled a more energy-efficient C–H activation pathway in the Zn-In-S system. While these findings offer theoretical guidance for the effective conversion of biomass utilizing semiconductor materials, the involvement of metal sulfide photocatalysts in biomass conversion, specifically in the activation of C-H bonds and the coupling of C-C bonds, has been extensively documented. It seems not to meet the criteria of high importance and urgency required for publication in Nature communication.

Some questions:

1. The yield of methylfuran (11 and 12) coupling product is very low. What are the other by-products?
2. Are other electron-rich heterocycles, such as thiophene, pyrrole, benzofuran, benzothiophene, compatible with this reaction system?
3. In Fig. 1, the author mentioned that sulfur radicals activate the benzylic C-H bond through the hydrogen atom transfer (HAT) pathway to generate free radical intermediates, which then undergo subsequent transformations. I believe this expression lacks precision, as nitrogen radicals, oxygen radicals, and halogen radicals can also accomplish this process. The author should consider making revisions and discussing this in the manuscript.
4. Some minor mistakes need correction. The abbreviation for the journal is incorrect in ref 7. Ensure consistency in the capitalization format of the first letters in the titles of cited references.

Reviewer #2 (Remarks to the Author):

This work reports Zn-In-S semiconductor as efficient catalyst for making coupling products from many biomass and fossil-derived chemicals. Moreover, combined experimental and computational investigations clarified a novel CPET mechanism as well as the rate-determining steps for the photocatalytic C-H activation, which are significantly different with the commonly accepted ET-PT mechanism. The experiments were designed in a very brilliant way while the computations fairly professional and in-depth. The results not only provide valuable insights into photocatalytic ET/PT processes, but also offer strategic options for analyzing the complicated photocatalysis. Therefore, I recommend its publication in the journal, albeit additional explanations on several issues are encouraged.

1. The idea of CPET mechanism is largely inspired by the KIE phenomenon over Zn-In-S, and the paper also mentioned that "The value above two is thus diagnostic of a primary KIE that is ascribed to CPET or PT-ET." While the KIE impact on the PT process (thus the rate-determining step) is agreed, is it also possible to show certain impact on the ET process?
2. Regarding Figure 3E, it is still hard to understand why CPET shows lower activation barrier than either the PT-ET or ET-PT pathway, since CPET seems conceptually to be in the middle of the two extremes of ET and PT situations.
3. How to treat the concentration as well as the partial reaction order of e-/h+ in the microkinetic analysis?
4. Further explanation on the sentence of "the adsorption energy of •PhCH₃⁺ cannot be calculated due to the artificial offset in the electrostatic potential reference under periodic boundary conditions," is encouraged. By the way, why not just compute the energy difference between two states of surface subtly adsorbed •PhCH₃⁺ and a desorbed one in the gas phase?

Leuven, March 22th, 2024

Subject: Response to the referees

Dear referees,

Please find enclosed the revised manuscript entitled: **‘Zinc-indium-sulfide favors efficient C–H bond activation by concerted proton-coupled electron transfer’** from authors Xuejiao Wu, Xueting Fan, Shunji Xie, Ivan Scodeller, Xiaojian Wen, Dario Vangestel, Jun Cheng, and Bert F. Sels.

We would like to thank you for your constructive comments on our manuscript. The addressed problems were processed with care, as much as we could. Accordingly, we have performed additional experimental work, incorporated relevant references, and did some rephrasing of sentences in order to avoid any confusion.

As a result of these changes and revisions, we are confident that our work is now strongly substantiated, effectively emphasizing the enhanced value derived from the peer review process.

You can find a point-by-point response to your comments below.

Sincerely yours,

Bert Sels

Full professor, Faculty of Bioscience Engineering, KU Leuven

Head of Sustainable Catalysis and Engineering, KU Leuven

Response to Reviewer 1

General Comments: Semiconductor photocatalysts play a vital role in the valorization of biomass by harnessing light energy to drive chemical reactions. In this manuscript, Cheng, Sels, and co-workers presented a Zn-In-S semiconductor catalyzed C–H bond activation strategy for the transformation of fossil and biomass derived chemicals. The authors carried out a mechanistic investigation to clarify the mechanism of zinc-indium-sulfides interfacial C–H bond activation process through isotope labeling, kinetic studies, and DFT calculations. Meanwhile, a microkinetic study has unveiled a more energy-efficient C–H activation pathway in the Zn-In-S system. While these findings offer theoretical guidance for the effective conversion of biomass utilizing semiconductor materials, the involvement of metal sulfide photocatalysts in biomass conversion, specifically in the activation of C-H bonds and the coupling of C-C bonds, has been extensively documented. It seems not to meet the criteria of high importance and urgency required for publication in Nature Communication.

Reply: Dear reviewer, we would like to express our gratitude for **your valuable and constructive feedback, which has contributed significantly to increasing the overall quality of our manuscript. Your recognition of our contributions to mechanistic studies is also highly appreciated.**

If we may, we must respectfully express our disagreement with your evaluation concerning the lack of significance and urgency in our paper. This perspective of yours seems to stem from the established use of metal sulfides in biomass valorization—a fact we are acutely aware of and have documented through the inclusion of the key references, encompassing both research articles and reviews, in our manuscript (*Ref. 3, 20, 21, 22, 23, 54*). Moreover, our team has contributed a review paper “*Metal Sulfide Photocatalysts for Lignocellulose Valorization*” (*Adv. Mater.*, 2021, 33, 2007129, *Ref. 3*), which reflects our significant dedication to this research area.

We have to emphasize that despite the growing application of metal sulfides in photocatalytic C–H bond activation for biomass valorization, **the detailed mechanisms** of such activation, which is the core of this manuscript, remain largely unexplored and elusive. **Our research provides a groundbreaking elucidation of the unexpected concerted proton-coupled electron transfer (CPET) mechanism as a pivotal factor in the superior activity of metal sulfides, addressing a critical gap in the existing knowledge of such semiconductor photocatalysis.** Therefore, **far from undermining the value of our work, the extensive use of metal sulfides in this domain actually underscores the importance of our findings.**

Generally, the mechanism of photocatalytic ‘element–H’ activations over semiconductors is described as dominated by a stepwise ET-PT, while **the more efficient concerted CPET pathway is rarely invoked or considered in the proposed reaction mechanism of semiconductors today.** It is important to note that the CPET mechanism has been extensively

investigated in model systems where the CPET reagents are either homogeneous metal complexes or organic bases. The popularity of this subject can be evidenced by several recently published reviewer papers, including *Chem. Rev.* 2022, 122, 1, *J. Am. Chem. Soc.* 2023, 145, 7050. This present contribution stands as the **first comprehensive report of** (to our knowledge) the **concerted CPET C–H activation mechanism over metal sulfides**. These facts highlight the substantial research opportunities and the critical importance of understanding the CPET mechanism in semiconductors, **stressing the urgent need for our findings to be recognized by the scientific community**.

In addition to the revelation of the concerted CPET mechanism over metal sulfide as the key reason contributing to their high activity for C–H activation, our study further provides unprecedented kinetic comprehension. A major hurdle in examining CPET kinetics on semiconductor materials is the intricate nature of their surface chemistry. To tackle this, we **propose an innovative approach that introduces the first microkinetic study** to rationalize the complex reaction network. This approach addresses the pivotal challenge in the kinetic study of interfacial proton-coupled electron transfer. The rate expressions derived from microkinetic analysis indicate several potential methods to increase the CPET activity. **Guided by this, we have modified the composition of the Zn-In-S, as recommended by our model, to enhance the rates of the rate-determining step** by increasing the driving force to further decrease the activation barrier.

The accordingly optimized catalyst was further evaluated in a wide scope of reactions. **Different from previous reports that predominantly focus on the self-coupling of one kind of chemical** (please see examples of *Angew. Chem., Int. Ed.* 2021, 60, 16399 and *Nat. Energy* 2019, 4, 575, which are cited as *Ref. 20* and *Ref. 54* in our manuscript), **our study included different chemicals and reaction types**. We explored substrates not only derived from biomass, such as furans and phenols, but also those originating from fossil-based aromatics.

Inspired by your suggestions, we have broadened our research scope by including **electron-rich thiophenes** in our study, a class of substrates less examined compared to aromatics and furans, thereby enriching the value of our work (please see **our replies to Comment 2**). Regarding reaction types, our study goes beyond C–C self-coupling, extending to C–C cross-coupling and C–N cross-coupling reactions. By incorporating a wide range of reactants and reaction types, our study underscores the versatility and promising future of metal sulfide-based CPET chemistry. We believe the publication of this work will significantly stimulate interest in CPET chemistry over semiconductors, which is vital yet underappreciated.

As demonstrated above, we firmly believe that our manuscript holds substantial novelty and importance, justifying its publication in *Nature Communications*.

Comment 1: The yield of methylfuran (11 and 12) coupling product is very low. What are the other by-products?

Reply and actions taken: We appreciate your observation regarding the lower yields obtained from methylfuran coupling. In our experiments, as illustrated in Fig. 5, a consistent reaction time of 14 hours was applied for processing all the different chemicals, including methylfurans. It was noted that the conversion rate of methylfurans was generally lower than that of the other chemicals, leading to their relatively modest yields due to incomplete conversion. To address this, we increased the reaction duration for methylfurans to 36 hours. This adjustment resulted in significantly improved yields, with 2-methylfuran showing a coupling product yield of 71%, and 2,5-dimethylfuran achieving 80%. It's important to note that the extended reaction time altered the selectivity, favoring the formation of trimers and tetramers (instead of dimers as previously observed). Side products from demethylation were also observed in noticeable amounts, more specifically, a 3 % yield of furan and 11% of methylfuran were obtained from the demethylation of 2-methylfuran and 2,5-dimethylfuran, respectively. We have incorporated these experimental details in the Supplementary Information (SI) (*SI, page 2, bottom two lines*) and updated *Fig. 5, Supplementary Table 4, Supplementary Fig. 34, and Supplementary Fig. 35* to reflect these changes.

Updated items: Fig. 5, Supplementary Table 4, Supplementary Fig. 34, and Supplementary Fig. 35.

Comment 2: Are other electron-rich heterocycles, such as thiophene, pyrrole, benzofuran, benzothiophene, compatible with this reaction system?

Reply and actions taken: We thank you for the constructive suggestion to increase the compatibility of our reaction system. Upon assessing the characteristics of the proposed substrates, it appears that our system may not efficiently convert pyrroles, benzofuran, and benzothiophene. Specifically, pyrroles, with their relatively low oxidation potential (please see *Synlett*, 2016, 27, 714-723), may not be conducive to the concerted CPET mechanism, as here ET reactions are much more favorable. Regarding benzofuran and benzothiophene, the coupling of these conjugated compounds can result in products with significant visible light absorption (please find absorption spectra of similar chemicals in *Can. J. Phys.* 2019, 97, 548), this characteristic could potentially impede light absorption or compromise stability under light irradiation. Nevertheless, we have tested representative chemicals for each category, including 2-methyl-pyrrole, benzofuran, and benzothiophene, as substrates, and as anticipated and in line with our study, these tests revealed low yields of coupling products of 7%, 1%, and 1%, respectively, after 24 hours of reaction.

In contrast, our studies showed that thiophenes are well-suited to our system. We further explored this by testing 2-ethyl thiophene, 3-ethyl thiophene, 2,5-dimethyl thiophene, and 2-methanol thiophene, achieving high coupling product yields of 91%, 92%, 71%, and 60%, respectively, after 24 hours of reaction. Notably, the primary by-products from the reactions involving 2,5-dimethyl thiophene and 2-methanol thiophene were 2-methyl-thiophene, resulting from the demethylation of 2,5-dimethyl thiophene, and thiophene-2-carboxaldehyde, formed through the oxidation of 2-methanol thiophene. These findings have been integrated into **Fig. 5** to showcase the viability of our system. We have incorporated additional experimental details in the Supplementary Information (SI) and updated **Fig. 5, Supplementary Table 4, and Supplementary Fig. 36-39** to reflect these changes. To further showcase the feasibility of our system toward thiophenes, a substrate category less explored in comparison to aromatics and furans, thus increasing the value of our system, we have added the following text to the main text:

Page 16, Paragraph: *“Additionally, the modified Zn₂In₂S₅ catalyst was utilized for the conversion of thiophenes, substrates that are less explored compared to aromatics and furans, and achieved high coupling product yields ranging from 60% to 92% (Fig. 5B, Supplementary Table 4, and Supplementary Fig. 36-39). This further underscores the versatility of the current CPET-based semiconductor system.”*

Updated items: **Fig. 5, Supplementary Table 4**

Added items: **Supplementary Fig. 36-39**

Comment 3: In Fig. 1, the author mentioned that sulfur radicals activate the benzylic C-H bond through the hydrogen atom transfer (HAT) pathway to generate free radical intermediates, which then undergo subsequent transformations. I believe this expression lacks precision, as nitrogen radicals, oxygen radicals, and halogen radicals can also accomplish this process. The author should consider making revisions and discussing this in the manuscript.

Reply and actions taken: We thank you for pointing out the lack of precision in this description. As suggested, we have modified **Fig. 1**, included two more references, and added the following text to the main text:

Page 3, Paragraph 2: *“Photocatalysis incorporating hydrogen atom transfer (HAT) has also been explored to get access to such benzylic/allylic radicals. Photo-excited organics, such as aromatic ketones and xanthene dyes, can function as HAT reagents to activate the C–H bonds directly.¹⁶ Alternatively, photo-redox catalysis drives the conversion of organics to corresponding radicals, e.g., thiols to sulfur and amines to nitrogen radicals, which could facilitate C–H bond activation. In this manner, photocatalysis engages in HAT through an indirect approach (Fig. 1, route 2).¹⁷”*

Modified figure: Fig. 1

Added references: Ref. 16 (Chem. Rev. 2022, 122, 1875) and Ref. 17 (Eur. J. Org. Chem. 2017, 2017, 2056)

Comment 4: Some minor mistakes need correction. The abbreviation for the journal is incorrect in ref 7. Ensure consistency in the capitalization format of the first letters in the titles of cited references.

Reply and actions taken: We thank you for pointing out these mistakes in reference formats. The journal abbreviation in ref. 7 has been corrected, and we have ensured uniformity in the capitalization format of the first letters in the titles of cited references. Additionally, we have carefully reviewed the format of all references to ensure compliance with the requirements of *Nature Communications*.

Response to Reviewer 2

General Comments: This work reports Zn-In-S semiconductor as efficient catalyst for making coupling products from many biomass and fossil-derived chemicals. Moreover, combined experimental and computational investigations clarified a novel CPET mechanism as well as the rate-determining steps for the photocatalytic C-H activation, which are significantly different with the commonly accepted ET-PT mechanism. The experiments were designed in a very brilliant way while the computations fairly professional and in-depth. The results not only provide valuable insights into photocatalytic ET/PT processes, but also offer strategic options for analyzing the complicated photocatalysis. Therefore, I recommend its publication in the journal, albeit additional explanations on several issues are encouraged.

Reply: Dear reviewer, **we highly appreciate your positive feedback on our work.** We are glad that our efforts to comprehend the intricate proton-coupled electron transfer chemistry on semiconductor surfaces have been acknowledged. Your remarks are important, which have played a crucial role in enhancing the precision and quality of our contribution. Our replies to the raised issues and the corresponding revisions are described as follows.

Comment 1: The idea of CPET mechanism is largely inspired by the KIE phenomenon over Zn-In-S, and the paper also mentioned that “The value above two is thus diagnostic of a primary KIE that is ascribed to CPET or PT-ET.” While the KIE impact on the PT process (thus the rate-determining step) is agreed, is it also possible to show a certain impact on the ET process?

Reply and actions taken: We thank you for reminding us of the potential contribution of the ET process to KIE. It is acknowledged that most ET processes typically lack KIE, though small isotope effects for pure ET have been observed in specific cases. In these cases, proton/deuterium reagents are not only the proton/deuterium acceptor/donor but also constitute the solvent fraction, thereby influencing vibrational modes and solvent-solute coupling (*J. Am. Chem. Soc.* 2012, 134, 16247-16254). Given that 10% toluene is utilized as a substrate in our study, the change of substrate from toluene to *d*8-toluene also results in a change in solvent composition, raising the possibility of a KIE contribution from ET. To address this pertinent concern, we conducted the reaction in deuterium CD₃CN and compared it with the reaction in CH₃CN, the result is included in a new Figure in SI (**Supplementary Fig. 9**), and the observed similarity in their activity effectively rules out the possibility of an ET contribution to the KIE. Therefore, we have added the following description to the main text:

Page 7, Paragraph 2: “In addition, similar activities were observed in the solvent of CH₃CN and CD₃CN (Supplementary Fig. 9), indicating the influence of solvent composition on the vibrational modes and solvent-solute coupling²⁸ only has a negligible contribution to KIE.”

Added references: Ref. 28 (J. Am. Chem. Soc. 2012, 134, 16247)

Added figure: Supplementary Fig. 9

Comment 2: Regarding Figure 3E, it is still hard to understand why CPET shows lower activation barrier than either the PT-ET or ET-PT pathway, since CPET seems conceptually to be in the middle of the two extremes of ET and PT situations.

Reply and actions taken: We thank you for pointing out the confusion in this part. We did not calculate the exact activation energy for these processes, theoretical studies of the concerted or sequential PCET kinetics still face great challenges because of the complexity of the reactions (please see for example, *Chem. Rev.* 2010, 110, 6939; *Energy Environ. Sci.*, 2012, 5, 7696; *Chem. Rev.* 2022, 122, 10599). Therefore, Figure 3E is only “conceptually”, as it was already indicated in the caption of Figure 3E, “*Conceptual illustration of the energetic advantages of the CPET process.*” The concerted pathway is kinetically favorable only when the sequential pathways, *i.e.*, ET-PT and PT-ET, are both highly energy-demanding, together with when large changes in p*K*_a and *E* are achieved upon one-electron oxidation or deprotonation, respectively (*J. Am. Chem. Soc.* 2021, 143, 560). We believe that this scenario is true for the benzylic C–H activation of toluene.

A new Supplementary figure (**Supplementary Fig. 10**) has been incorporated to elucidate the thermodynamics for toluene benzylic C–H oxidation. For the stepwise pathways, the high oxidation potential (*E* = 2.26 V) poses challenges for driving the ET-PT pathway, and toluene is a very weak acid (p*K*_a in CH₃CN = 41), requiring a strong base for the PT-ET to take place.

Additionally, a large change in pK_a ($\Delta pK_a = 54$) and E ($\Delta E = 3.18$ V) was achieved upon one-electron oxidation or deprotonation, respectively. In other words, the second step for both sequential pathways, that are deprotonation of toluene cation radical (PT in ET-PT pathway, $pK_a = -13$) and oxidation of toluene anion radical (ET in PT-ET pathway, $E = -0.926$), is highly thermodynamically favorable. Thus, the rate-determining step with the exothermic event in the sequential pathway through the concerted approach results in a favorable driving force. The much more favorable driving forces are accompanied by reduced activation barriers, facilitating a more rapid concerted CPET process for toluene oxidation compared to sequential transfer pathways. Even though we cannot provide the exact data of lower activation energy for the CPET step and the PT-ET step, The favorable CPET process is rationalized through the thermodynamic analysis and supported by the different product features from the photocatalytic conversion of deuterated toluene- d_8 over $ZnIn_2S_4$ and P25 (Please see from Page 9, Paragraph 1). To make our points clearer and to avoid confusion, we have modified the following sentences into the main text:

Page 9, Paragraph 1: “For benzylic C–H activation in toluene, the high 2.26 V oxidation potential renders the ET-PT pathway difficult to drive, and toluene is a very weak acid (pK_a in $CH_3CN = 41$), requiring a strong base for the PT-ET pathway. On the other hand, deprotonation of the toluene cation radical (PT in ET-PT pathway, $pK_a = -13$) and oxidation of toluene anion radical (ET in PT-ET pathway, $E = -0.926$), the second step for both sequential pathways, are highly thermodynamically favorable (Supplementary Fig. 10).¹³ Thus, coupling the favorable event (second step) through the concerted pathway is then able to compensate for unfavorable energetics (first step). The much more favorable driving forces in the concerted approach are often accompanied by low activation barriers, allowing concerted CPET to proceed more rapidly than competing sequential transfer pathways (Fig. 3E).⁹”

Added figure: Supplementary Fig. 10

Comment 3: How to treat the concentration as well as the partial reaction order of e^-/h^+ in the microkinetic analysis?

Reply and actions taken: We thank you for kindly reminding us to also include the discussion on concentration as well as the e^-/h^+ in the microkinetic analysis. To answer your question we performed the photocatalytic conversion of toluene with different concentrations (2.5 vol%, 5 vol%, 7.5 vol%, 10 vol%, and 20 vol%) over $Zn_2In_2S_5$ and P25, and these results are included in two new Supplementary figures (**Supplementary Fig. 14** and **Supplementary Fig. 20**). As shown in **Supplementary Fig. 14**, the activity of $Zn_2In_2S_5$ exhibits a linear increase from 2.5 vol% to 10 vol%, with a slight deceleration from 10 vol% to 20 vol%. In the case of A4 as RDS, a two-order dependency on the toluene concentration is expected. The results confirmed that A2 (following eq. 2) is the RDS instead of A4 (following eq. 3). The first-order dependency

also indicates that $K_{Ad1} \times c_{RC-H} + K_{Ad2} \times c_{RC}$, in denominator of eq. 2 is much smaller than 1, indicating a small adsorption energy, which is consistent with our calculation results.

About the partial reaction order of e^-/h^+ in the microkinetic analysis, we believe here the reviewer refers to eq. 6 (B3 as RDS for TiO₂) in which h_s^+ is present in both the numerator and denominator of the equation. However, similar to that for Zn₂In₂S₅, the activity of TiO₂ (P25) increases linearly from 2.5 vol% to 10 vol%, and then slows down a bit from 10 vol% to 20 vol% (**Supplementary Fig. 20**). As analyzed above, this indicates that the denominator part including h_s^+ is negligible and can be ignored. Therefore, in the case of TiO₂, the activity follows a pseudo-first-order rate dependency on h_s^+ .

The following sentences were then added to the main text to include the discussion on the concentration and h_s^+ in microkinetic analysis:

Last Paragraph, Page 12: “Additionally, the activity of Zn₂In₂S₅ demonstrates a linear increase with the rise in toluene concentration from 2.5 vol% to 10 vol%, with a slight deceleration from 10 vol% to 20 vol% (Supplementary Fig. 14). With A4 as RDS, a two-order dependency on the toluene concentration would be expected. The observed first-order dependency confirms that A2 determines the rate. According to eq. 2, under the condition of small adsorption energy (Supplementary Table 3), a linear relation of activity to toluene concentration is anticipated.”

Page 14, Paragraph 4: “The activity of P25 increases almost linearly with the concentration of toluene to around 20 vol% (Supplementary Fig. 20). This indicates that the denominator part including “ c_{RC-H} ” is negligible and can be ignored, implying a first-order dependency on h_s^+ for both eq. 5 and eq. 6.”

Added figure: Supplementary Fig. 14, Supplementary Fig. 20

Comment 4: Further explanation on the sentence of “the adsorption energy of $\bullet\text{PhCH}_3^+$ cannot be calculated due to the artificial offset in the electrostatic potential reference under periodic boundary conditions,” is encouraged. By the way, why not just compute the energy difference between two states of surface subtly adsorbed $\bullet\text{PhCH}_3^+$ and a desorbed one in the gas phase?

Reply and actions taken: We thank the reviewer for this comment. The adsorption energy (ΔE_{ad}) of $\bullet\text{PhCH}_3^+$ is usually calculated using the following equation:

$$\Delta E_{ad} = E(\text{slab} - \bullet\text{PhCH}_3^+) - E(\text{slab}) - E(\bullet\text{PhCH}_3^+)$$

where $E(\text{slab} - \bullet\text{PhCH}_3^+)$, $E(\text{slab})$, and $E(\bullet\text{PhCH}_3^+)$ denote the total energies of the $\bullet\text{PhCH}_3^+$ adsorbed slab, the bare slab and the $\bullet\text{PhCH}_3^+$ in the gas phase, respectively.

Due to periodic boundary conditions applied in the modeling, the reference (zero) of the electrostatic potential in such a setup is artificial and has no physical meaning (please see for example, *J. Chem. Phys.* 2009, 131, 154504; *Phys. Rev. B* 1981, 24, 7412; *J. Chem. Theory*

Comput. 2010, 6, 880). The addition or removal of the $\bullet\text{PhCH}_3^+$ in the model changes the net charge of the cell and therefore the electrostatic potential reference. Total energies from these calculations are hence useless, and cannot be used to determine relative energies.

For the calculation method advised by the reviewer:

$$\Delta E_{\text{ad}} = E(\text{slab} - \bullet\text{PhCH}_3^+) - E(\text{slab} + \bullet\text{PhCH}_3^+)$$

where $E(\text{slab} + \bullet\text{PhCH}_3^+)$ denotes the total energy of the system that contains the bare slab and the desorbed $\bullet\text{PhCH}_3^+$ in the gas phase. We agree with the reviewer that the energy difference calculated using this method may be useful as the net charge of the cell remains the same. However, the interaction between $\bullet\text{PhCH}_3^+$ and the surface cannot be neglected unless $\bullet\text{PhCH}_3^+$ is far enough from the surface. Thus, the width of the vacuum must be large, which may significantly increase the computation costs. Moreover, it is not allowed to select a specific charge state of the $\bullet\text{PhCH}_3^+$ in the calculation, and the hole is found more likely to localize on the slab rather than on the PhCH_3 in the gas phase. Therefore, calculating the adsorption energy using this method is impractical, and, to the best of our knowledge, no previous work has succeeded in doing so. The following text together with three new *Ref.* have been added to the main text to clarify the situation:

Page 13, the Last Paragraph: “Unfortunately, the adsorption energy of $\bullet\text{PhCH}_3^+$ cannot be calculated. This is because periodic boundary conditions are applied in the modeling, the reference (zero) of the electrostatic potential in such a setup is artificial and has no physical meaning.⁴⁴⁻⁴⁶ Addition or removal of the $\bullet\text{PhCH}_3^+$ in the model changes the net charge of the cell and therefore also the electrostatic potential reference. Total energies from these calculations are hence useless, and cannot be used to determine relative energies. However, the high reported energy of $\bullet\text{PhCH}_3^+$ ⁴⁷ indicates a strong desorption energy.”

Added references: Ref. 44-46 (*J. Chem. Phys.* 2009, 131, 154504; *Phys. Rev. B* 1981, 24, 7412; *J. Chem. Theory Comput.* 2010, 6, 880)

REVIEWERS' COMMENTS

Reviewer #1 (Remarks to the Author):

The author highlights the significance of the CPET mechanism in the C-H activation process of semiconductor catalysts. Reviewers concur with the author's perspective, finding it intriguing as it offers valuable insights into achieving C-H functionalization in semiconductor photocatalysis. In my opinion, the benzylic C-H bonds have relatively low bond dissociation energy (BDE), and their functionalization has been extensively researched. I don't believe that the homocoupling reaction of benzylic C-H bonds presents significant synthetic challenges. Furthermore, semiconductor Zn-In-S, acting as a photocatalyst, has also been investigated to some extent in the realm of C-H activation. Furthermore, compared to other heterogeneous and homogeneous catalytic systems (such as *Adv. Synth. Catal.* 2018, 360, 932-941; *Nat. Commun.* 2023, 14, 6366), there are notable limitations in substrate scope. It is only applicable to the homocoupling or functionalization of benzylic C-H bonds of electron-rich aromatic hydrocarbons, thus constraining its practical utility and the chemoselectivity of heterocyclic substrates is not good. In conclusion, I recommend submitting this manuscript to an alternative journal.

Reviewer #2 (Remarks to the Author):

The author has made significant modifications to the manuscript and clarified many previous concerns. The novel CPET mechanism, excellent catalytic performance, as well as the rate-determining steps analysis for the photocatalytic C-H activation, has been rationalized by combined experimental characterizations and DFT calculations. Therefore, I recommend its publication in this journal as it is.

Response to Reviewer 1

General Comments: The author highlights the significance of the CPET mechanism in the C-H activation process of semiconductor catalysts. Reviewers concur with the author's perspective, finding it intriguing as it offers valuable insights into achieving C-H functionalization in semiconductor photocatalysis. In my opinion, the benzylic C-H bonds have relatively low bond dissociation energy (BDE), and their functionalization has been extensively researched. I don't believe that the homocoupling reaction of benzylic C-H bonds presents significant synthetic challenges. Furthermore, semiconductor Zn-In-S, acting as a photocatalyst, has also been investigated to some extent in the realm of C-H activation. Furthermore, compared to other heterogeneous and homogeneous catalytic systems (such as *Adv. Synth. Catal.* 2018, 360, 932-941; *Nat. Commun.* 2023, 14, 6366), there are notable limitations in substrate scope. It is only applicable to the homocoupling or functionalization of benzylic C-H bonds of electron-rich aromatic hydrocarbons, thus constraining its practical utility and the chemoselectivity of heterocyclic substrates is not good. In conclusion, I recommend submitting this manuscript to an alternative journal.

Reply: We would like to express our gratitude for your recognition of the contributions our study makes to mechanistic insights, which stand as the key achievements of our research. We also appreciate your efforts in reviewing our paper, which has helped to enhance the quality of our manuscript.

Regarding your evaluation of lacking significance of our paper, we believe we have already presented a very comprehensive and strong sustained argument in our last round of rebuttals, showing novel insights into the C-H activation mechanism of the semiconductor photocatalysis. Somewhat unexpectedly, CPET came out as the mechanism. We maintain that our publication will significantly stimulate the interest in CPET chemistry over such semiconductors, an area that is vital, yet highly underappreciated today. We firmly believe that our manuscript holds substantial novelty and importance, justifying its publication in the prestigious journal *Nature Communications*.

Response to Reviewer 2

General Comments: The author has made significant modifications to the manuscript and clarified many previous concerns. The novel CPET mechanism, excellent catalytic performance, as well as the rate-determining steps analysis for the photocatalytic C-H activation, has been rationalized by combined experimental characterizations and DFT calculations. Therefore, I recommend its publication in this journal as it is.

Reply: We greatly appreciate your positive feedback on our work and thank you for reviewing our manuscript. Your comments have been invaluable in helping us enhance our manuscript.